# Fully Automated Bioreactor-Based pH-Cycling System for Demineralization: A Comparative Study with a Conventional Method

**DOI:** 10.3390/ma16144929

**Published:** 2023-07-10

**Authors:** Ryouichi Satou, Naoko Miki, Miyu Iwasaki, Naoki Sugihara

**Affiliations:** Department of Epidemiology and Public Health, Tokyo Dental College, Tokyo 101-0061, Japan; naokom@tdc.ac.jp (N.M.); iwasakimiyu@tdc.ac.jp (M.I.); sugihara@tdc.ac.jp (N.S.)

**Keywords:** pH-cycling, enamel, microhardness, demineralization, preventive dentistry

## Abstract

This study aimed to develop an automated pH-cycling system using inexpensive commercial components that can replicate pH fluctuations in the oral cavity and salivary clearance to compare demineralization characteristics with the conventional method. The study found that the newly developed cycle-1 group showed improved demineralization properties, including apparent lesion depth, surface roughness, Vickers hardness, mineral loss, and depth of demineralization, compared to the control group. Additionally, the cycle-2 group, which had a longer cycle interval, showed further improvements in the demineralization properties. This system can replicate the differences in dental damage caused by differences in meals, snacking frequencies, and lifestyle rhythms, making it useful in cariology, preventive dentistry research, and oral care product development. It can be constructed using inexpensive commercial products, significantly reducing research costs and improving reproducibility and fairness between different experimental facilities. The system can replicate lifestyle rhythms, such as meals, sleep, and oral clearance by saliva, making it an in vitro pseudo-oral cavity.

## 1. Introduction

In cariology research, many studies have utilized an in vitro experimental method using bovine teeth and a demineralizing solution to produce dental caries with excellent reproducibility [1,2,3,4]. However, the preparation of demineralizing solutions varies across different experimental systems and researchers, leading to non-standardization even within universities in the same country. Moreover, several in vitro demineralization/remineralization experiments have failed to replicate the natural pH changes occurring in the oral cavity, and instead measured artificially induced demineralization, which poses a significant concern [5]. In the oral cavity, continuous and rapid pH fluctuations occur multiple times within a short period after each food and drink consumption, and the solution in the oral cavity is constantly replaced by swallowing [6,7,8]. Unless an experimental system that can faithfully replicate these biological changes is established, it would be difficult to accurately investigate the effects of the target components or products on demineralization/remineralization.

The pH-cycling model, which was developed to replicate pH changes in the oral cavity caused by food and drink consumption, has been widely accepted and utilized in the scientific and dental industries as an appropriate alternative method for investigating the caries prevention effects of fluoride application in humans and animals [5,9]. Several pH-cycling models, including manual and automated models, have been developed for both caries and erosive lesions [10,11,12,13]. Automation through microcontroller control has been particularly effective in addressing issues such as forgetting to exchange solutions and preparation errors and also reduced operator errors [5,10,14]. Matsuda et al. developed an automated pH-cycling device capable of replicating continuous pH variation represented by a Stephan curve, and established an experimental system capable of incorporating temporal demineralization and remineralization in samples [14,15]. This system was designed so that the fluid in a specimen-containing beaker could achieve the minimum pH 4.5 for demineralization and the maximum pH 6.8 for remineralization [14]. This device provides high reproducibility and accuracy while minimizing operator errors through automation when evaluating the effect of fluoride on caries dynamics [14,15]. However, these devices are custom-made, making it challenging for other researchers or companies to conduct similar experiments on the same experimental system. In addition, there is no pH cycling system that is standardized and commonly used in laboratories and companies worldwide.

Therefore, we aimed to develop a “fully automated pH-cycling system” that can be easily constructed by combining commercially available products and can install control programs on a PC, even for individuals without specialized knowledge. We hypothesized that developing a pH-cycling device that can be easily implemented would be useful for many cariology researchers and those involved in the sale and development of oral care products as it would accelerate research and allow for standardization of experimental conditions in different research laboratories by exchanging the control program. The null hypothesis of this study is that demineralization with this system is no different than with conventional methods, so there is no point in conducting experiments with the system. Our study aimed to develop a fully automated pH-cycling system and a control program by combining commercially available products, and to compare the demineralization methods of our system with conventional non-continuous demineralization methods and different pH profiles measured by histology, cariology, and dental engineering.

## 2. Materials and Methods

### 2.1. Construction of a Fully Automated pH-Cycling Acid Challenge System Using a Bioreactor

Figure 1 shows a schematic diagram of the fully automated pH-cycling system developed in this study, which includes a control PC, a pH controller, three peristaltic pumps, and a bioreactor vessel (Figure 1a). Each component and its current price as of 2023 are detailed in Table 1. The device’s core structure uses a commercially available radial flow type bioreactor system (BC-200cc, Biott Corp., Tokyo, Japan) and control software (Control program ver2.1, Biott Corp., Tokyo, Japan), which has been optimized for pH-cycling experiments. Specific modifications to the software involved expanding the number of input programs for pH control from 20–100 to enable the reproduction of fine pH fluctuations in a short period of time. The system can control up to eight units using a single PC and software, and multiple experiments can be conducted simultaneously by expanding the pH controller and bioreactor vessels. Figure 1b shows the detailed structure of the bioreactor vessel, featuring ports on its lid for solutions, gases, and various sensors and a stirring wing at its center for magnetic stirrers. The upper part of the vessel was composed of stainless steel and silicone packing, while the lower part was composed of reinforced glass, which allowed autoclaving sterilization. To summarize the device’s pH control method, the set value of the pH controller was monitored and adjusted in real time using the control PC. If the pH value obtained by the electrode submerged in the solution inside the vessel fell outside the dead zone of the set value, the motor of the peristaltic pump for demineralization and remineralization was activated to inject the solution into the vessel. In the interval method, the motor speed of the peristaltic pump was constant, and the amount of solution delivered was controlled only by the operating time. In other words, if the pH value obtained by the submerged electrode in the solution deviated from the set value, the motor continued to deliver the solution without rest time; however, as the pH value approached the set value, the rest time increased to limit the amount of solution delivered. A photograph of the system used in this study is shown in Figure 1c. The vessel was installed in an incubator and operated at 37 °C, which replicated the conditions inside the human body. Other parts were placed outside the incubator to prevent overheating and facilitate the easy replenishment of the solution (Figure 1c).

The pH-cycling profile used in this study is shown in Figure 2a. The Stephan curve was programmed according to Matsuda et al. [11,14,15]. In each cycle, the duration of a pH value of 5.5 or lower was set to an average of 40 ± 2 min, the recovery duration from a pH value of 5.5–7.3 was set to 20 ± 3 min, and the total duration from the start of the cycle to the return to the initial pH value of 7.3 was set to an average of 60 ± 5 min. The acid solution tank contained a demineralization solution (0.2 M lactic acid buffer solution, Ca:1.5 mM, P:0.9 mM, pH 4.5, DS:5.5), while the alkali solution tank contained a remineralization solution (0.02 M HEPES-based buffer solution, Ca:1.5 mM, P:0.9 mM, pH 7.3, DS:5.5) [16]. It is recommended to prepare the remineralization solution immediately right before the experiment because its composition changes gradually from the moment the calcium chloride solution is added, followed by the start of the precipitation reaction.

In this study, 3 experimental groups were created to compare the developed system with the conventional method of discontinuous demineralization, with variations in the presence of cycling and intervals between each cycle. The program content for each experimental group is shown in Figure 2b. The control group replicated the conventional method and underwent demineralization for 10 h, followed by 20 h of remineralization. The cycle-1 group aimed to measure the effect of the presence of cycles and underwent 15 cycles with an interval of 60 min. The total demineralization and remineralization times in the cycle-1 group were similar to those in the control group (demineralization time: 40 min × 15 cycles; remineralization time: [recovery time of 20 min + cycle interval of 60 min] × 15 cycles). The cycle-2 group, with a cycle interval of 120 min, was created to compare the effect of the cycle interval on the demineralization properties min, as shown in Figure 2b.

### 2.2. Preparation of Enamel Samples

A total of 27 bovine anterior mandibular teeth were used. Enamel blocks measuring 1 cm × 1 cm × 1 cm were then prepared and polished to a mirror finish using water-resistant abrasive papers of #1000, #2000, and #4000 grit.

### 2.3. Three-Dimensional Laser Microscopic Observation

Following wax removal, the samples were dehydrated using an ascending ethanol series. To measure the step height profile between the experimental (ES) and reference (RS) surfaces after pH-cycling, a three-dimensional (3D) measurement laser microscope (LEXT OLS4000, Olympus Corp., Tokyo, Japan) was utilized. The extent of tooth defects resulting from the acid challenge was noted. The measurement area was 645 µm × 645 µm, with photographs taken at the boundary between the acid-demineralized ES and wax-protected RS. Finally, 3D measurements were taken at 5 locations for each sample, and the mean and standard deviation values were calculated. 

### 2.4. Micro-Vickers Hardness Measurement

After dehydration, the micro-Vickers hardness of the samples was measured using a hardness tester (HMV-1; Shimadzu Corp., Tokyo, Japan). The Vickers hardness (HV) values were measured with an indentation load and time of 0.49 N and 20 s, respectively. To account for individual sample differences, changes in HV values before and after the experiment (∆HV = RS − ES) were calculated. The HV and ∆HV values were recorded at 5 locations per sample, and the mean ∆ standard deviation was calculated.

### 2.5. Cross-Sectional Morphology through Scanning Electron Microscopy 

Following pH-cycling, each sample was rinsed with xylene, and carbon vapor deposition was applied to the surface of the analyte sample. The tooth surfaces were examined under a scanning electron microscope (SU6600; HITACHI Ltd., Tokyo, Japan) at an accelerating voltage of 15 kV. The samples were then embedded in a polyester resin (Rigolac, Nisshin EM, Tokyo, Japan) to create polished sections, and the cross-sections were observed.

### 2.6. Contact Microradiography (CMR) 

The imaging conditions and analysis method were based on those by Sato et al., with reference to Angmar’s formula [16,17]. The samples were embedded in a polyester resin (Rigolac, Nisshin EM, Tokyo, Japan) to prepare 100 µm thick polished sections. Soft X-ray imaging was performed (CMR-3, Softex, Tokyo, Japan) with a 20 µm Ni filter and light microscopy at 200× magnification using a glass plate (High Precision Photo Plate, HRP-SN-2; Konica Minolta, Tokyo, Japan). Imaging was performed using a tube voltage, tube current, and radiation time of 15 kV, 3 mA, and 15 min, respectively. The resulting images were analyzed using the Image Pro Plus software (version 6.2; Media Cybernetics Inc., Silver Spring, MD, USA) and image analysis system (HC-2500/OL; OLYMPUS Corp., Tokyo, Japan) to obtain the concentration profile. Mineral loss value (ΔZ) and lesion depth (Ld) were determined to compare the extent of demineralization. ΔZ was calculated using a formula, and Ld was defined as the distance from the enamel surface to the lesion location with a mineral content higher than 95% compared to that of sound enamel.

### 2.7. Statistical Analysis

Mean values and standard deviations were calculated for the 9 samples to compare the effectiveness of the 3 programs. The significance of the results was determined using the Kruskal–Wallis one-way analysis of variance with a threshold value of *p* < 0.01. Post hoc comparisons were performed using the Bonferroni test. Data were analyzed and graphs were prepared using Origin software (ORIGIN 2023, Lightstone Corp., Tokyo, Japan). 

## 3. Results

### 3.1. Step Height Profiles Measured by 3D Laser Microscopylaser Microscopy after pH-Cycling

Figure 3 shows the 3D laser microscopy measurements of the surface profiles after pH-cycling. The left side of Figure 3a–c shows the reference surface (RS), which was not demineralized and was protected with wax, whereas the right side shows the demineralized ES. In the control group, ES was significantly demineralized, and a 37.933 ± 3.164 μm defect was observed on the enamel surface (Figure 3a). In the cycle-1 group, the difference in height between RS and ES decreased to 32.283 ± 1.594 μm, and a significant inhibition of demineralization was observed compared to the control group (*p* = 0.0024) (Figure 3b,d). The cycle-2 group had an even smaller height difference of 17.674 ± 1.374 μm compared to the of cycle-1 group, and the least amount of enamel demineralization was observed in the cycle-2 group among the three groups (*p* = 3.840 × 10^−7^) (Figure 3c,d). 

### 3.2. Calculated Average Roughness after pH-Cycling

Figure 4 represents the calculated average surface roughness (Sa) for each group. Figure 4a–c represents the variations in surface elevation in 3D and cross-sectional plots along the X- and Y-axes, precisely at the position of the white triangle. Figure 4d shows a box plot of the results of each group, indicating the mean value with a white square, median with a horizontal line, lower quartile as the lower limit, and upper quartile as the upper limit. The control group exhibited significant irregularities on the enamel surface, with a mean Sa value of 2.924 ± 0.721 μm and a median of 2.893 μm (2.381–3.681), representing the largest roughness among all the groups (Figure 4a,d). The cycle-1 group had a significant reduction in Sa value of 0.515 ± 0.079 μm and a median of 0.513 μm (0.461–0.564) compared to the control group (*p* < 0.001) (Figure 4b,d). The cycle-2 group exhibited the least Sa value among the three groups, with a mean value of 0.382 ± 0.073 μm and a median of 0.389 μm (range: 0.324–0.432); however, no significant difference was observed compared to the cycle-1 group (Figure 4c,d). The Sa of the cycle-2 group also exhibited a significant decrease compared to the control group (*p* < 0.001). The enamel Sa values of the negative control group, which did not undergo pH-cycling, had a mean value of 0.034 ± 0.009 μm and a median value of 0.035 (0.028–0.036) (Figure 4d).

### 3.3. Micro-Vickers Hardness and Its Changes after pH-Cycling

Figure 5 shows the results of the Vickers microhardness test for the demineralized surfaces in each experimental group. The control group had a mean microhardness value of 21.448 ± 2.249 and a median value of 21.098 (20.497–22.914), which were significantly lower than those of the cycle-1 and cycle-2 groups (*p* < 0.01, Figure 5a). The cycle-1 group had a mean and median microhardness of 74.572 ± 6.484 and 75.462 (68.743–80.336), respectively. The cycle-2 group had the highest mean and median values of 123.193 ± 13.832 and 127.054 (110.094–129.447), respectively, among all the groups, which were significantly different from those of the cycle-1 group (*p* < 0.01, Figure 5a).

The changes in the Vickers microhardness before and after pH-cycling are shown in Figure 3b. The mean and median values of the control group were 239.847 ± 10.571 and 292.660 (291.166–302.809), respectively, which were the largest among the three groups and were significantly higher than the values of the cycle-1 and cycle-2 groups (*p* < 0.01, Figure 5b). The mean and median values of the cycle-1 group were 221.979 ± 5.719 and 224.255 (219.801–225.216), respectively, indicating a decrease. The mean and median values of the cycle-2 group were 162.919 ± 23.311 and 169.947 (138.531–178.059), respectively, which were the least among all the groups and were significantly different from of the values of the cycle-1 group (*p* < 0.01, Figure 5a).

### 3.4. Cross-Sectional Scanning Electron Microscope Observations after pH-Cycling

Figure 6 shows backscattered electron images of the cross sections perpendicular to the demineralized surface after pH-cycling. In the control group, a decrease in signal intensity and an increase in enamel prism interprismatic spaces were observed within the range of 0–50 μm from the surface, particularly with severe demineralization and collapse of the enamel prism structure at a depth of 0–30 μm below the surface (Figure 4a,d). The cycle-1 group exhibited a narrow range of enamel prism structure disappearance and widened inter-prismatic spaces of demineralization, limited to a depth of 0–15 μm from the surface, with no significant decrease in signal intensity observed beyond 20 μm (Figure 4b,e). The cycle-2 group also exhibited similar demineralized areas and widened interprismatic spaces, limited to a depth of approximately 0–15 μm, but no significant decrease in signal intensity was observed beyond 20 μm from the surface, with a signal intensity comparable to that of the sound area (Figure 4c,f).

### 3.5. Measurement of Mineral Loss Value and Lesion Depth by Contact Microradiography Analysis

Figure 7 shows a CMR image of a longitudinal section of the enamel after pH-cycling and a graph demonstrating variations in mineral content (vol% μm) with respect to the depth for each group. In the control group, a region with low signal intensity was present in the surface layer of the dental enamel at a depth of 25–50 μm, and a rise in the curve was observed at around 50 μm (Figure 7a,d). In the cycle-1 group, a rise in the curve was at a shallow depth of 0–25 μm from the surface, with a region where mineral content was recovered from a relatively shallow depth observed (Figure 7b,d). In the cycle-2 group, mineral content reached over 80% in the 0–25 μm range, with the highest mineral content among the three groups observed at the shallowest depth (Figure 7c,d).

Figure 8 shows the mineral loss (ΔZ, vol% μm) and lesion depth (Ld, μm) values of each group analyzed using CMR. The ΔZ value of the control group was 16,150.780 ± 1478.721 vol% μm, which was significantly greater than all the other groups (*p* < 0.01, Figure 8a). The cycle-1 group showed a reduction in ΔZ value to approximately two-thirds of that observed in the control group, with a value of 9848.587 ± 518.646 vol% μm. The cycle-2 group had the least ΔZ value of approximately half of that of the control group, with a value of 7140.258 ± 408.690 vol% μm. A significant difference was observed between the cycle-2 and cycle-1 group (*p =* 0.004; Figure 8a). The Ld value was largest in the control group at 147.694 ± 7.039 μm, with a significant difference observed between the values of the control, Cycle-1, and Cycle-2 groups (*p* < 0.01, Figure 8b). The Ld value of the cycle-1 group decreased to 111.319 ± 11.972 μm compared to that of the control group and further decreased to 107.069 ± 10.373 μm in the cycle-2 group. However, there was no significant difference between the cycle-1 and cycle-2 groups (*p =* 0.822; Figure 8b).

## 4. Discussion

### 4.1. Comparison of Demineralization Characteristics between the Conventional Method and Fully Automated Bioreactor-Based pH-Cycling Method

The control group, which used the conventional method without pH-cycling, showed a demineralized surface with substantial loss of enamel prism extending from the surface to a depth of 0–50 μm, along with increased Sa due to the roughness of the demineralized surface (Figure 4 and Figure 6). In contrast, using the newly developed pH-cycling device in the cycle-1 group led to an improvement in all aspects, including the amount of substantial loss, surface roughness, micro-Vickers hardness, mineral loss, and lesion depth of demineralization, despite the total time for demineralization and remineralization being equal to that of the control group (Figure 3, Figure 4, Figure 5, Figure 6, Figure 7 and Figure 8). In the cycle-2 group, the demineralization properties were further reduced compared to those in the cycle-1 group by extending the cycling interval, exhibiting a healthy enamel-like appearance (Figure 3, Figure 4, Figure 5, Figure 6, Figure 7 and Figure 8). The human oral cavity undergoes simultaneous pH recovery, remineralization, and demineralization of the buffering action and clearance of saliva [6,18]. Previous studies using pH-cycling for early caries have been conducted, but they only involved soaking in demineralization and remineralization solutions for a certain period of time without replicating the temporal pH changes observed in the oral cavity [3,19]. Moreover, an environment in which the balance between demineralization and remineralization (as in the control group) is skewed for a long period of time is unlikely to occur in the actual oral cavity and is considered an extreme model, such as plaque adhering to the deep parts of high-risk areas for caries and occlusal fissures [3,20,21]. Furthermore, as reflected by the large surface roughness of the control group, the differences in demineralization patterns depending on the site of rapid demineralization can cause significant variability in measurements at each measurement site in the scanning electron microscopy or CMR sections, affecting the experimental accuracy. Compared to the control group, the cycle-1 group underwent short periods of demineralization and remineralization similar to the actual oral cavity environment before and after meals. Therefore, both demineralization and remineralization progress continuously and mildly, resulting in an improvement in demineralization properties such as surface roughness and mineral loss, and a uniform demineralization pattern can be expected. A comparison between the cycle-1 and cycle-2 groups demonstrated that extending the cycling interval improved the demineralization status, and the use of this new pH-cycling system suggests that differences in tooth damage due to meal or snack frequencies and lifestyle rhythms can be reproduced. A lifestyle with frequent snacking or consumption of food and drinks before bedtime can increase the risk of dental caries [8,21]. This study suggests that the system programmability of a PC allows for the input of various lifestyle patterns, thereby creating samples with different demineralization characteristics reflecting the differences in these patterns. Matsuda et al. developed a custom automatic pH-cycling device that can replicate pH changes in the oral cavity and constructed an experimental system to investigate temporal changes in the demineralization of dental hard tissues caused by continuous pH-cycling [14,15]. Previous studies using an automatic pH-cycling device on enamel have reported a decrease in the demineralization rate and an increase in mineral loss and depth of demineralization with an increasing number of cycles. Similar trends were observed in our study [10,11,14,15]. The system in our study is deemed useful as an in vitro pseudo-oral cavity capable of replicating daily rhythms, such as eating and sleeping, through programmatic inputs and oral clearance via saliva. This equipment has extensive applicability in cariology and preventive dentistry.

### 4.2. Advantages and Prospects of the Fully Automated Bioreactor-Based pH-Cycling System

Our device, constructed using a combination of commercially available devices and software that can be easily acquired and cost-effectively replaced (Table 1), is markedly more affordable than previous pH-cycling devices, which were custom-made by researchers and involved the small-scale production of parts commissioned by companies, costing between $50,000–$100,000. The entire system can be prepared for approximately $9300, which reduces the experimental cost by one-fifth to one-tenth (Table 1). Nonetheless, the exchange of protocols between research facilities that use different equipment is difficult and, even with the same equipment, issues arise with reproducing experimental environments due to changes in configurations of parts due to wear and replacement. However, the system used in this study monitors pH changes in real time through a pH controller connected to a PC via a USB, which controls the ON/OFF timing of the motor automatically, enabling the creation of the same pH profile regardless of the type of pump or configuration (Figure 1). The operating conditions and timing of the pump used to create a pH profile can be saved as a file (PC extension: STP) and shared with other researchers via an email attachment or by uploading to a cloud system, as long as the OS is Windows, and the file name is “program name.STP.” Experiments on demineralization and remineralization are highly dependent on the experience of the researchers, technique used, and the potential for operational errors that could affect the experimental results. However, automating the experiments using this device improved the reproducibility and reduced the difficulty of the experiments. Regardless of the experimental skills or experience of the researcher, the ability to reproduce experiments in different research facilities can significantly improve the reproducibility and fairness of the research. Additionally, a PC-controlled programmatic approach has other benefits. The pH value for demineralization in this system is based on the reference pH value of 5.5 for the enamel of permanent teeth. However, the critical pH values for deciduous teeth and dentin are 5.7–6.2 and 6.0–6.2, respectively [10,22]. By adjusting the composition of the demineralization solution and program, this model can be modified to create acid erosion models, root surface caries, and enamel caries depending on the target sample. Due to its low viscosity, the demineralization solution of this system would be particularly useful as a model for dental erosion caused by citric and acetic acids present in beverages. While the incidence of dental erosion and root caries is increasing, the lack of established in vitro experimental models hinders the development of preventive measures [23,24]. However, the use of this system can advance the development of fluoride applications and preventive foods for dental erosion and root caries. Additionally, this system enables the evaluation of the performance of oral care products, which was previously carried out independently by manufacturers, with comparable and fair data. This system is considered to contribute to the research and development of the dental industry and product development.

Our system’s limitation is its inability to use gels for demineralization and remineralization because it can only operate on liquids. When quantifying the mineral loss and lesion depth of demineralization using micro-radiography, it is desirable for the sample to reproduce the white spot and have a subsurface demineralization image without actual enamel loss [14,25,26,27]. When creating subsurface demineralization images of enamel, a high-viscosity demineralization solution is commonly used, and it is typically applied mildly for a long time to simplify the experiment. In previous studies, gels have been prepared using thickening agents such as 2% HEC or 8% CMC in a demineralization solution [3,4,19,20,28,29]. However, if the solution is made into a gel, the diffusion of calcium and phosphate ions released from the tooth during surface acid demineralization is slower than that in the liquid state, resulting in a higher ion saturation level near the surface and a lower demineralization rate. Gel-based demineralization has also been used as a model for artificial plaque-induced demineralization in microbiological experiments [3,4,19]. The purpose of the microbiological experimental model was to reproduce the state in which plaque adheres to the tooth enamel by increasing the viscosity of the descaling solution and to replicate the ion dynamics at the interface between the plaque and the sample. However, using high-viscosity gels, such as 2% HEC with our system, causes delays in reaching the target pH, even when the pump is set to its maximum speed, making it impossible to reproduce the Stephan curve according to the program. The high load on the pump required to move high-viscosity gels creates a significant reduction in the durability of the system, making it unsafe. Therefore, our system cannot use gels and can only reproduce a demineralization-like state using liquids. Although this system is optimal for reproducing caries with cavitation caused by acid or modeling acid erosion caused by liquids, it is considered unsuitable for reproducing white spot lesions due to plaques. While this issue may be resolved using a pump with sufficient torque, conventional methods may be more appropriate when using gels. Furthermore, it is necessary to investigate the demineralization characteristics when using extracted human teeth in addition to bovine teeth.

## 5. Conclusions

In the cycle-1 group, which employed the newly developed fully automated bioreactor-based pH-cycling system, all demineralization characteristics, including substantial loss, surface roughness, micro-Vickers hardness, mineral loss, and lesion depth, were improved despite its equivalent total duration of demineralization and remineralization with the control group. In the cycle-2 group, which extended the cycling intervals beyond those of cycle-1, further improvements in the demineralization characteristics were observed. These findings suggest that this system can reproduce differences in tooth enamel damage caused by meal or snack frequencies and the rhythm of daily life. This system is also cost-effective because it can be constructed entirely from commercially available products. Furthermore, the transfer of experimental programs between researchers is possible and complete automation allows for improved reproducibility and fairness across different research facilities, regardless of the experimenter’s skill or experience. This system is an excellent in vitro pseudo-oral cavity system that can reproduce daily rhythms, such as eating and sleeping, as well as oral clearance by saliva. This system is expected to contribute to research on cariology and preventive dentistry, and to the development of the dental industry and product development.

## Figures and Tables

**Figure 1 materials-16-04929-f001:**
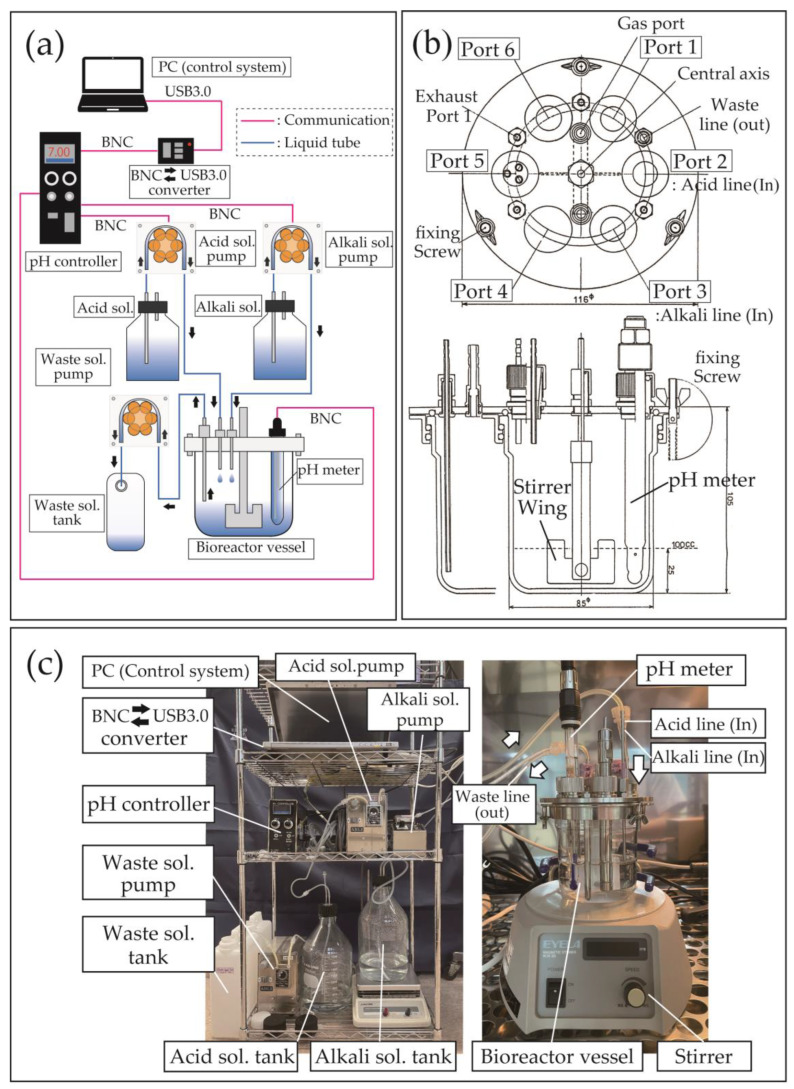
Design and components of a fully automated pH-cycling system. (**a**) Overview of the fully automated pH-cycling system. The red lines indicate communication cables, and the blue lines indicate solution tubes. (**b**) Design drawing of the bioreactor vessel. (**c**) Image and part arrangement of the fully automated pH-cycling system developed in this study. The bioreactor vessel is operated inside the incubator, while the other parts are arranged outside the incubator. Arrows indicate the direction of solution flow.

**Figure 2 materials-16-04929-f002:**
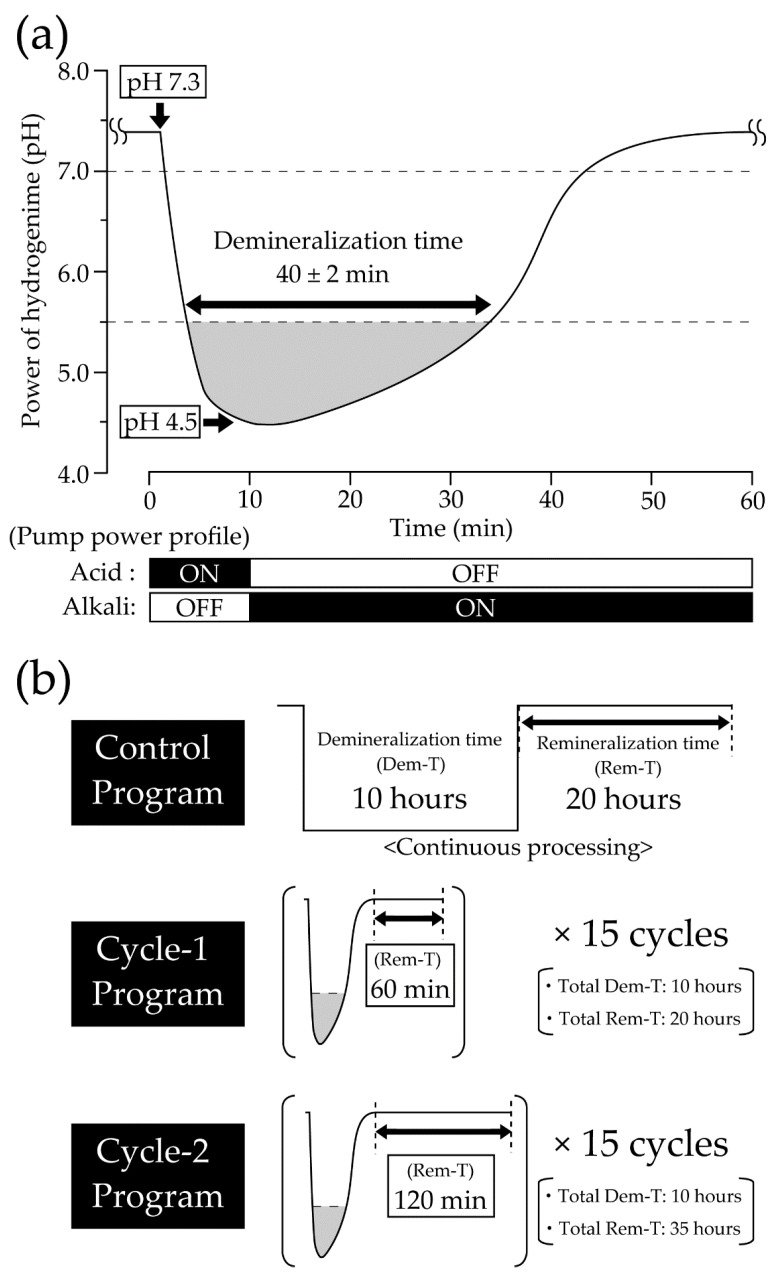
Overview of the programmed temporal pH changes. (**a**) Overview of the programmed temporal pH changes in the pH-cycling system. The time below the critical pH of the enamel (pH 5.5) is considered the demineralization time and is set to 40 ± 2 min. (**b**) The experimental group programs were as follows: the control group replicated the conventional method, with demineralization for 10 h followed by remineralization for 20 h; the cycle-1 group performed 15 cycles with an interval of 60 min; the total demineralization and remineralization times for the cycle-1 group were similar to those for the control group; the cycle-2 group performed 15 cycles with an interval of 120 min.

**Figure 3 materials-16-04929-f003:**
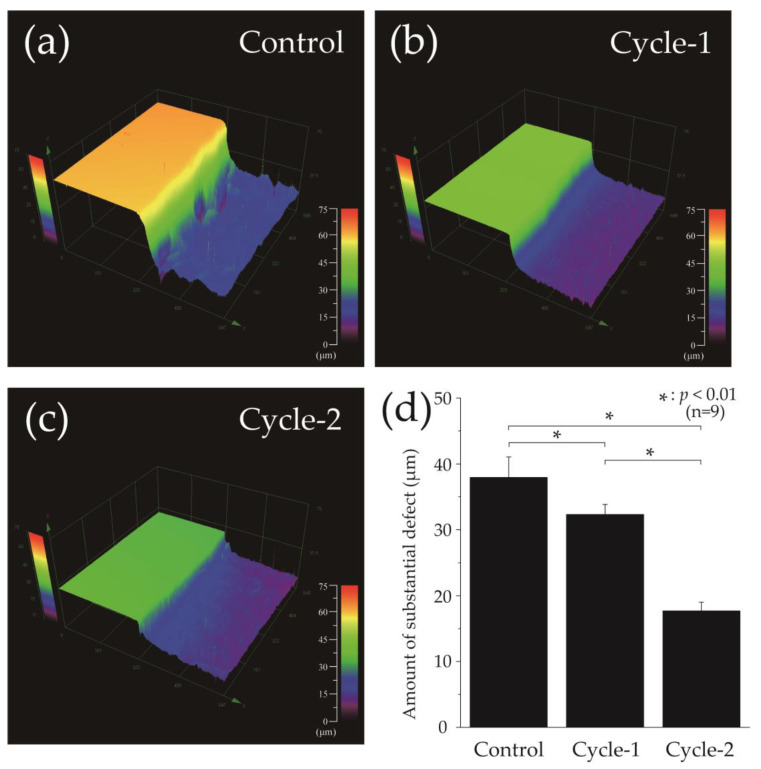
Height difference profiles measured using a 3D laser microscope. Boundary images of the reference and experimental surfaces after pH-cycling of the (**a**) control, (**b**) cycle-1, and (**c**) cycle-2 groups. The left side of figures (**a**–**c**) shows the RS protected by wax and not demineralized; the right side shows the ES that has been demineralized. (**d**) Graphical representation of the substantial defects due to demineralization (n = among the three groups (n = 9, * *p* < 0.01).

**Figure 4 materials-16-04929-f004:**
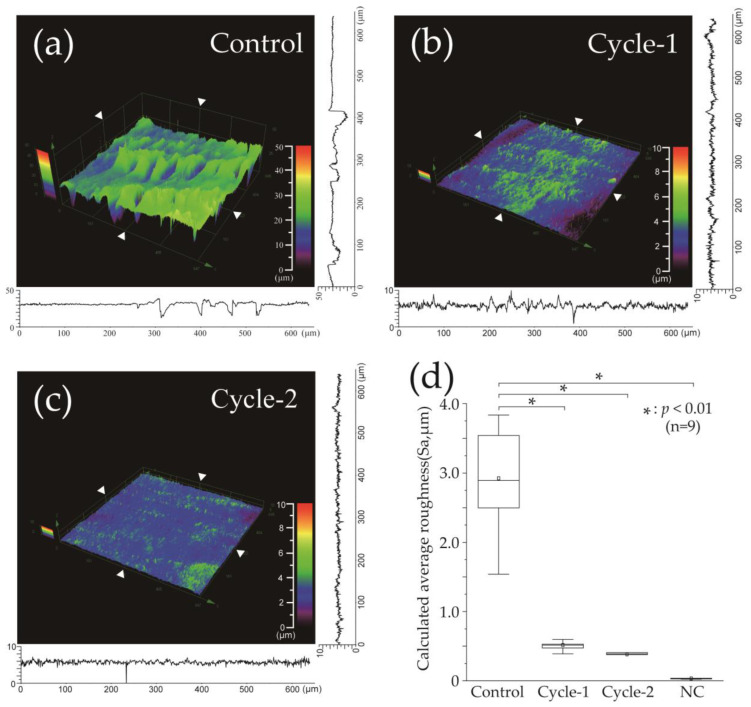
Calculated average roughness after pH-cycling. 3D images of the surface roughness after the pH-cycling of the (**a**) control, (**b**) cycle-1, and (**c**) cycle-2 groups. The horizontal and vertical line roughness graphs are shown on the X and Y axes, respectively, at the positions of the white triangles in the image. (**d**) The distribution of the dataset can be visualized using the boxplot (n = 9, * *p* < 0.001). The median values are indicated by a horizontal line in the middle of the box, and the lower and upper boundaries indicate the 25th and 75th percentiles, respectively. White squares indicate mean values.

**Figure 5 materials-16-04929-f005:**
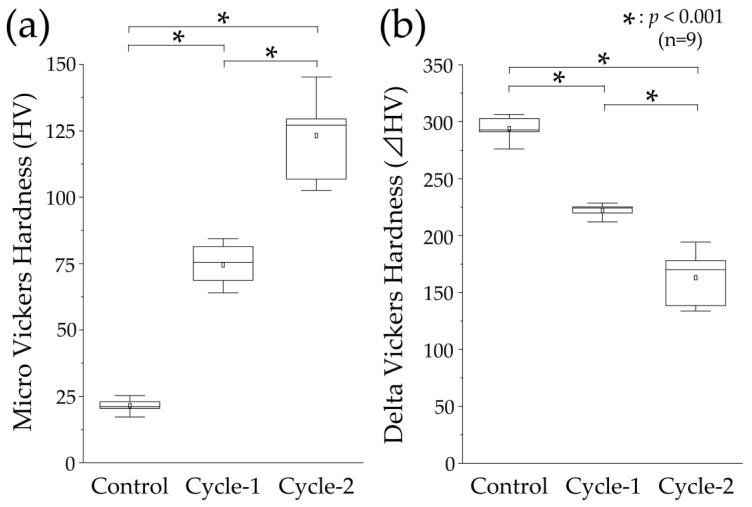
Micro-Vickers hardness measurements. (**a**) Boxplot of micro-Vickers hardness (HV) values after pH-cycling (n = 9, * *p* < 0.001). The median value is indicated by the horizontal line in the middle of the box and the lower and upper boundaries indicate the 25th and 75th percentiles, respectively. The white squares indicate the mean value. (**b**) Boxplot of ΔHV values (difference in the HV values between the RS and ES) (n = 9, * *p* < 0.001).

**Figure 6 materials-16-04929-f006:**
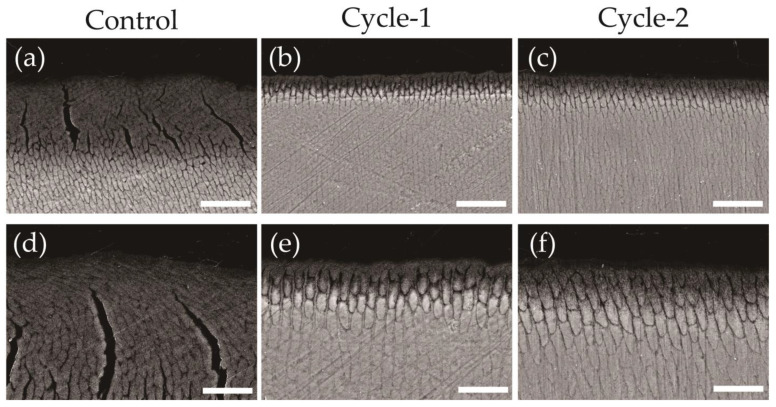
Scanning electron microscope (SEM) images of enamel cross-sections after pH-cycling. Cross-sectional SEM image of the control (**a**,**d**), cycle-1 (**b**,**e**), and cycle-2 (**c**,**f**) groups. (**a**–**c**) Scale bar is 25 μm. All images were recorded at 1000-fold magnification carbon deposition sample. (**d**–**f**) Scale bar is 12.5 μm. All images were recorded at 2000-fold magnification, carbon deposition sample.

**Figure 7 materials-16-04929-f007:**
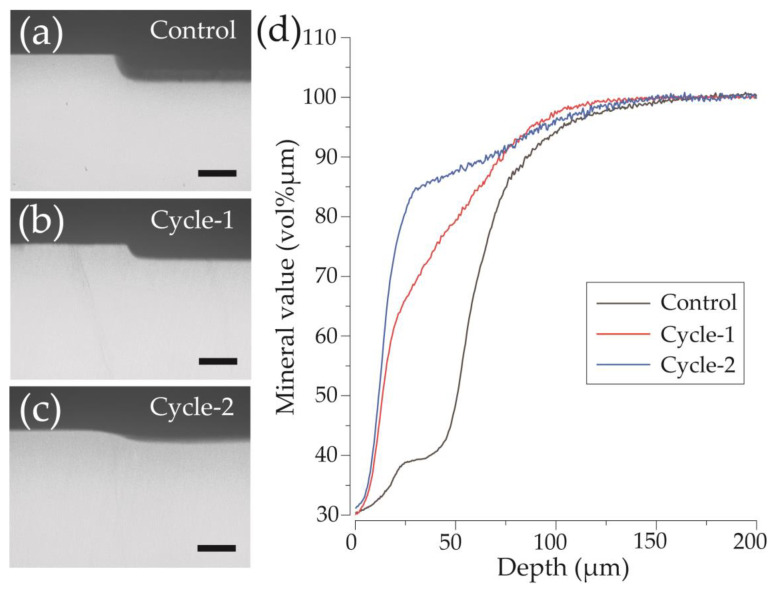
Contact microradiography (CMR) images of enamel cross-sections after pH-cycling. CMR cross-sectional images of the reference and experimental surfaces after the acid challenge of the (**a**) control, (**b**) cycle-1, and (**c**) cycle-2 groups. Scale bar is 50 μm. (**d**) Graphical representation of the mineral value by tooth depth. Black, red, and blue lines show the control, cycle-1, and cycle-2 groups, respectively.

**Figure 8 materials-16-04929-f008:**
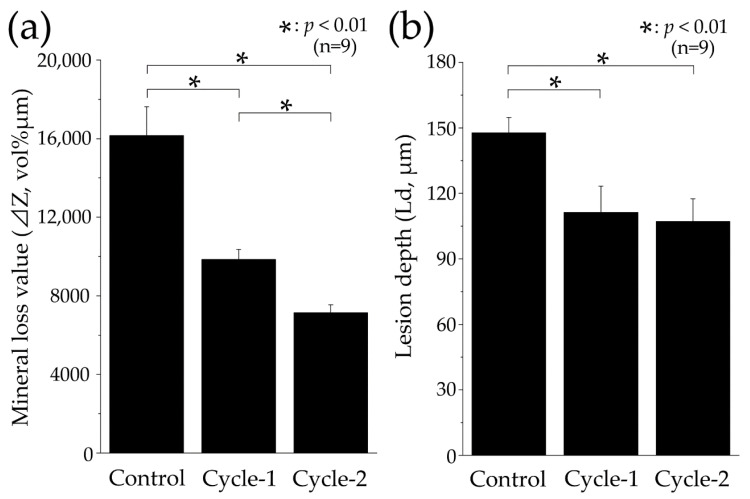
Graphical representation of mineral loss (ΔZ) and lesion depth (Ld) values after pH-cycling. (**a**) Graphical representation of mineral loss value (ΔZ, n = 9, * *p* < 0.01). All nine samples were measured, and the mean ± SD is shown. (**b**) Graphical representation of lesion depth value (Ld, n = 9, * *p* < 0.01). From the surface prior to the demineralization experiment, the depth of demineralization was determined up to a site showing 95% healthy enamel. All 9 samples are measured, and the mean ± SD is shown.

**Table 1 materials-16-04929-t001:** List of components and prices for pH-cycling system.

No	Parts Name	Quantity	Price (dollar)
1	Bioreactor vessel	1	2100.00
	BC-200cc, Biott Corp., Tokyo, Japan		
2	pH electrode sensor	1	530.00
	EASYFERM PLUS PHI S8 120, HAMILTON Corp., Bonaduz, Switzerland		
3	pH electrode cable	1	150.00
	Length 1.5 m, BNC connector		
4	pH controller	1	1200.00
	DJ-1023, Biott Corp., Tokyo, Japan		
	Control method: Time-division proportion or Interval		
	Measurement range: pH2.00–12.00		
	Communication method: RS485		
	Output: 2 AC outlets		
5	Acid solution pump	1	870.00
	AC-2110II, ATTO Corp., Tokyo, Japan		
	Flow rate: 5 to 1500 mL/h (using tubes with inner diameters of 1–3 mm)		
6	Alkali solution pump	1	870.00
	AC-2110II, ATTO Corp., Tokyo, Japan		
	Flow rate: 5 to 1500 mL/h (using tubes with inner diameters of 1–3 mm)		
7	Waste solution pump	1	870.00
	AC-2110II, ATTO Corp., Tokyo, Japan		
	Flow rate: 5 to 1500 mL/h (using tubes with inner diameters of 1–3 mm)		
8	Magnetic stirrer	1	320.00
	HS-30DN, ASONE Corp., Tokyo, Japan		
9	Communication cable	1	135.00
10	pH cycling control software set (for Windows OS)	1	1970.00
	Control program ver2.1, Biott Corp., Tokyo, Japan		
	Data display program ver4.0, Biott Corp., Tokyo, Japan		
	GRAPH2 ver4.3, Biott Corp., Tokyo, Japan		
11	USB3.0-RS485 converter	1	270.00
	USB3.0-RS485 converter, Biott Corp., Tokyo, Japan		
	Required for connecting Windows PC and pH controller		
		Total Price:	9285.00

## Data Availability

All data are included in the manuscript.

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
