# Peer review of "Fully Automated Bioreactor-Based pH-Cycling System for Demineralization: A Comparative Study with a Conventional Method"

_materials, 2023, doi:10.3390/ma16144929_

Round 1

Reviewer 1 Report

Thanks Authors to choose MDP and Materials to publish their manuscript

This study aimed to develop an automated pH-cycling system using inexpensive commercial components, thanks to technology and multidisciplinary fields machinery like this can greatly help the field of research in the areas of our interest.

introduction fully centers the topic.

Materials and methods and results are well dexterous and clear, they also present excellent iconography.

Discussion are ok.

Perhaps a limitation of this study is the sample size, which could have been larger than 27 units, and future studies even on teeth that were precetely extracted and prepared for systematics could have been used.

Author Response

<Reviewer 1>

Thanks Authors to choose MDP and Materials to publish their manuscript

This study aimed to develop an automated pH-cycling system using inexpensive commercial components, thanks to technology and multidisciplinary fields machinery like this can greatly help the field of research in the areas of our interest.

introduction fully centers the topic.

Materials and methods and results are well dexterous and clear, they also present excellent iconography.

Discussion are ok.

> We strongly appreciate the reviewer's comment. We are thankful for the time and energy you expended.

Perhaps a limitation of this study is the sample size, which could have been larger than 27 units, and future studies even on teeth that were precetely extracted and prepared for systematics could have been used.

> The reviewer's comment is correct. It is possible that the sample size may exceed the number used in the current experiment. In such cases, it is possible to increase the number of pumps and vessels to accommodate the larger sample size. Up to 8 systems can be simultaneously controlled with a single control device. Further details have been provided in the following section. For more information, please see the following section of the paper.

Page 2, Line 80-82

The system can control up to eight units using a single PC and software, and multiple experiments can be conducted simultaneously by expanding the pH controller and bioreactor vessels.

In accordance with the reviewer's comment, we have added discussion part as described below.

Page 15, Line 435-437

Furthermore, it is necessary to investigate the demineralization characteristics when using extracted human teeth, in addition to bovine teeth.

Reviewer 2 Report

The main question addressed by the research is the preparation of an alternative and affordable device for the formation of artificial caries.

It is important to standardize the formation of artificial carious lesions in the field of cariology. A new technical equipment developed in this field may be of interest to the readers. I have a few concerns which should be addressed.

Please give null hypothesis at the and of Introduction section. And discuss it.

Please add the rationale behind the study in Introduction Section. Why do we need this type of setup? Clarify.

The subject is more of a niche study of methodology in the field of cariology. The gap it fills is limited.

Introducing an advanced equipment that can be prepared with an affordable budget adds to the subject area compared with other published material.

The conclusions are consistent with the evidence and arguments presented and they address the main question posed

The references are appropriate

Tables and figures are sufficiently rich and informative. It does not require any improvement.

Author Response

<Reviewer 2>

The main question addressed by the research is the preparation of an alternative and affordable device for the formation of artificial caries.

It is important to standardize the formation of artificial carious lesions in the field of cariology. A new technical equipment developed in this field may be of interest to the readers. I have a few concerns which should be addressed.

> We strongly appreciate the reviewer's comment. We are thankful for the time and energy you expended.

Please give null hypothesis at the and of Introduction section. And discuss it.

> In accordance with the reviewer's comment, we have added introduction part as described below.

Page 2, Line 62-64

The null hypothesis of this study is that demineralization with this system is no different than with conventional methods, so there is no point in conducting experiments with the system.

Please add the rationale behind the study in Introduction Section. Why do we need this type of setup? Clarify.

> We appreciate the reviewer's comment on this point. In accordance with the reviewer's comment, we have added introduction part as described below.

Page 2, Line 54-55

In addition, there is no pH cycling system that is standardized and commonly used in laboratories and companies worldwide.

The subject is more of a niche study of methodology in the field of cariology. The gap it fills is limited.

>Thank you for understanding our opinion. This system caters to researchers and companies in the oral care product industry, making it a niche area of focus. However, conducting experiments with pH-cycling is essential for assessing the performance of newly developed technologies and products. Therefore, we believe that there is significant latent demand for it.

Introducing an advanced equipment that can be prepared with an affordable budget adds to the subject area compared with other published material.

>The reviewer's comment is correct. Comparison of this system with other devices is necessary. In accordance with the reviewer's comment, we have added introduction part as described below.

Page 2, Line 47-50

This system was designed such that the fluid in a specimen-containing beaker could achieve the minimum pH 4.5 for demineralization and the maximum pH 6.8 for remineralization [14].

The conclusions are consistent with the evidence and arguments presented and they address the main question posed

The references are appropriate

Tables and figures are sufficiently rich and informative. It does not require any improvement.

> We wish to express our appreciation to the reviewers for their insightful comments on our paper.

Reviewer 3 Report

This is a systematic and logically well-structured manuscript with interesting research material. A pH cycling system is a meaningful study useful for laboratory research including dental caries.

In the introductory section, the purpose of the study is clearly described along with the justification and hypothesis of the study.

In the method section, reliability and validity of research design and research procedures are secured.

In the results section, the derived results are highly readable. However, since this study is an experimental study and the number of samples is small, it is necessary to design the alpha = 0.01 level to be more challenging to verify the significance of the research results in Figures 3, 4, and 8. It is recommended to present the p-value as it is.  

In the discussion section, the research results are well discussed with previous studies. In addition, the limitations of the study were presented and the strengths of this study were clearly presented.

Author Response

<Reviewer 3>

This is a systematic and logically well-structured manuscript with interesting research material. A pH cycling system is a meaningful study useful for laboratory research including dental caries.

>We wish to express our appreciation to the reviewers for their insightful comments on our paper. The comments have helped us significantly improve the paper.

In the introductory section, the purpose of the study is clearly described along with the justification and hypothesis of the study.

In the method section, reliability and validity of research design and research procedures are secured.

> We strongly appreciate the reviewer's comment. We are thankful for the time and energy you expended.

In the results section, the derived results are highly readable. However, since this study is an experimental study and the number of samples is small, it is necessary to design the alpha = 0.01 level to be more challenging to verify the significance of the research results in Figures 3, 4, and 8. It is recommended to present the p-value as it is. 

>In accordance with the reviewer's comment, we have changed sentence as described below.

We have corrected the resulting p-values.

Page7, Line 203-209

In the cycle-1 group, the difference in height between RS and ES decreased to 32.283 ± 1.594 μm, and a significant inhibition of demineralization was observed compared to that of the control group (p = 0.0024) (Figures 3b and 3d). The cycle-2 group had an even smaller height difference of 17.674 ± 1.374 μm compared to that in the of cycle-1 group, and the least amount of enamel demineralization was observed in cycle-2 group among the three groups (p =3.840×10^-7) (Figures 3c and 3d).

>The p-value in Figure 4 is less than 0.0001, so the text and figure were corrected to p < 0.001.

Page 12, Line 307-308

A significant difference was observed between the cycle-2 and cycle-1 group (p = 0.004; Figure 8a).

Page 12, Line 312-313

However, there was no significant difference between the cycle-1 and cycle-2 (p = 0.822; Figure 8b).

In the discussion section, the research results are well discussed with previous studies. In addition, the limitations of the study were presented and the strengths of this study were clearly presented.

>Again, thank you for giving us the opportunity to strengthen our manuscript with your valuable comments and queries. We have worked hard to incorporate your feedback and hope that these revisions persuade you to accept our submission.

Reviewer 4 Report

With interest I’ve read the paper “Fully Automated Bioreactor-based pH-Cycling System for

Demineralization: A Comparative Study with a Conventional Method”.

The authors sought to develop an automated pH-cycling system using inexpensive commercial components that can replicate natural oral pH fluctuations. The study is thoroughly presented, but the main limitation is a small sample size and a short time of probiotic intake. The study is of great interest to a range of readers among dental researchers in the field of caries and erosion. However, some comments should be addressed.

Introduction

I suggest to describe previously reported models in more details to underline the advantages of the proposed one.

Methods

The methods used in this study are appropriate and the section is well-written.

Line 214 “Figure 4 shows represents” typo, remove one verb.

Line 220 “a mean Sa value of 2.924 ± 0.721 μm and a median of 2.924 μm (2.381–3.681)” were means and median really the same up to the third decimal (it seems from the boxplot that there were not)?

Discussion

I think that it would be interesting to discuss the possibility of replicating subsurface demineralization. Although the methods used in the study mainly assessed surface characteristics, one of the most challenging parts of caries modeling is replicating a lesion which is predominantly subsurface, while surface lesion mainly replicates acid erosion.

Minor spelling check is required.

Author Response

<Reviewer 4>

With interest I’ve read the paper “Fully Automated Bioreactor-based pH-Cycling System for Demineralization: A Comparative Study with a Conventional Method”.

The authors sought to develop an automated pH-cycling system using inexpensive commercial components that can replicate natural oral pH fluctuations. The study is thoroughly presented, but the main limitation is a small sample size and a short time of probiotic intake. The study is of great interest to a range of readers among dental researchers in the field of caries and erosion. However, some comments should be addressed.

> We wish to express our appreciation to the reviewers for their insightful comments on our paper. The comments have helped us significantly improve the paper. Our responses to the referees’ comments are as follow:

The study is thoroughly presented, but the main limitation is a small sample size and a short time of probiotic intake.

>The reviewer's comment is correct. We found that similar previous studies have been performed with n=5 to 15 (CMR and height difference profiles measured using a 3D laser microscope). We started with n=12 samples per group. However, handling 100μm thick samples for CMR was difficult and 1-3 samples in each group were broken. Therefore, the final number of samples was n=9.

>In addition, the following section describes how to improve the sample size.

Page 2, Line 80-82

The system can control up to eight units using a single PC and software, and multiple experiments can be conducted simultaneously by expanding the pH controller and bioreactor vessels.

Introduction

I suggest to describe previously reported models in more details to underline the advantages of the proposed one.

> Thank you very much for providing important comments. In accordance with the reviewer's comment, we have added following sentence in introduction section.

Page 2, Line 47-50

This system was designed such that the fluid in a specimen-containing beaker could achieve the minimum pH 4.5 for demineralization and the maximum pH 6.8 for remineralization [14].

Methods

The methods used in this study are appropriate and the section is well-written.

>We wish to thank the reviewer for this comment.

Line 214 “Figure 4 shows represents” typo, remove one verb.

>The reviewer's comment is correct. We deleted the “shows” and corrected the sentence (Page 8, Line 217).

Line 220 “a mean Sa value of 2.924 ± 0.721 μm and a median of 2.924 μm (2.381–3.681)” were means and median really the same up to the third decimal (it seems from the boxplot that there were not)?

>The reviewer's comment is correct. We thank the reviewer for this comment. The median was incorrect and has been corrected.

Page 8, Line 222-224

The control group exhibited significant irregularities on the enamel surface, with a mean Sa value of 2.924 ± 0.721 μm and a median of 2.893 μm (2.381–3.681), representing the largest roughness among all the groups (Figures 4a and 4d).

Discussion

I think that it would be interesting to discuss the possibility of replicating subsurface demineralization. Although the methods used in the study mainly assessed surface characteristics, one of the most challenging parts of caries modeling is replicating a lesion which is predominantly subsurface, while surface lesion mainly replicates acid erosion.

>Thank you very much for providing important comments. The present cross-sectional SEM image was able to reproduce the demineralization that exists sub-surface (Figure 6). The change in Vickers hardness also proves that the system was able to reproduce the structural decay inside the tooth (Figure 5). Although gel cannot be used in this system, it is thought that subsurface demineralization can also be created.

Again, thank you for giving us the opportunity to strengthen our manuscript with your valuable comments and queries. We have worked hard to incorporate your feedback and hope that these revisions persuade you to accept our submission.